# Effects of Exogenous α-Naphthaleneacetic Acid and 24-Epibrassinolide on Fruit Size and Assimilate Metabolism-Related Sugars and Enzyme Activities in Giant Pumpkin

**DOI:** 10.3390/ijms232113157

**Published:** 2022-10-29

**Authors:** Chen Chen, Xuan-Min Wu, Liu Pan, Ya-Ting Yang, Hai-Bo Dai, Bing Hua, Min-Min Miao, Zhi-Ping Zhang

**Affiliations:** 1College of Horticulture and Plant Protection, Yangzhou University, Yangzhou 225009, China; 2Joint International Research Laboratory of Agriculture and Agri-Product Safety of Ministry of Education of China, Yangzhou University, Yangzhou 225009, China; 3Key Laboratory of Plant Functional Genomics of the Ministry of Education, Jiangsu Key Laboratory of Crop Genomics and Molecular Breeding, Yangzhou University, Yangzhou 225009, China

**Keywords:** giant pumpkin, α-naphthaleneacetic acid, 24-epibrassinolide, source capacity, photoassimilate, sugar transport, sink strength, fruit size

## Abstract

Size is the most important quality attribute of giant pumpkin fruit. Different concentrations and application frequencies of α-naphthaleneacetic acid (NAA) and 24-epibrassinolide (EBR) were sprayed on the leaves and fruits of giant pumpkin at different growth stages to determine their effects and the mechanism responsible for fruit size increase. NAA+EBR application improved source strength, and further analysis indicated that NAA+EBR markedly boosted net photosynthetic rate (Pn), stomatal conductance (Gs), transpiration rate (Tr) and the expression level and activity of galactitol synthetase (GolS), raffinose synthetase (RS), and stachyose synthetase (STS), resulting in an increase in the synthesis of photoassimilate, especially stachyose. Concomitantly, NAA+EBR spray increased stachyose and sucrose contents throughout pumpkin fruit growth and the concentrations of glucose and fructose at 0 and 20 days post-anthesis (DPA) in peduncle phloem sap, implying that such treatment improved the efficiency of assimilate transport from the peduncle to the fruit. Furthermore, it improved the expression and activity of alkaline α-galactosidase (AGA), facilitating assimilate unloading, providing carbon skeletons and energy for fruit growth, and increasing fruit weight by more than 44.1%. Therefore, exogenous NAA and EBR increased source capacity, transportation efficiency, and sink strength, overall promoting the synthesis and distribution of photoassimilate, ultimately increasing fruit size.

## 1. Introduction

Giant pumpkins (*Cucurbita maxima*, Cucurbitaceae) are a special type of pumpkin with a very large single fruit, often weighing hundreds of kilograms [1,2]. Its peel is smooth and colorful, mainly red, yellow, or orange-yellow; and the flesh is mostly yellow or orange. It is edible and has great ornamental value. Giant pumpkin is a selection from the variety “Atlantic giant” patented by Howard in 1979 [3], and an annual contest is organized annually around the world to reward the heaviest giant pumpkin. Additionally, owing to the directional breeding of varieties and recent improvements in cultivation technology, the single-fruit world record weight of giant pumpkins is constantly being updated [2,4]. In China, owing to the recent development of agricultural tourism and traditional agriculture, the cultivation of giant pumpkins has increased in various regions and has become a key project attracting tourists to many agricultural parks across the country.

Giant pumpkin fruit-size remains the most important fruit quality attribute and is of great significance for searching and developing practical strategies for improvement [5]. Researchers have long studied how to successfully grow giant pumpkins weighing hundreds of kilograms. Growth cycle duration is a major factor affecting giant pumpkin size [2,6]. The growth period of normal pumpkins is approximately 90–120 d, while the giant pumpkins generally take 140–160 d to grow, being sown in early May and harvested in late September to increase the fruit weight [7]. However, not all regions are suitable for the long-season, over-summer cultivation of giant pumpkins; for example, pumpkins cannot survive in the hot summer areas of southern China. Giant pumpkin fruit-size can also be increased by reducing planting density, increasing leaf area, improving leaf nutrient supply, and expanding the nutrient absorption area using adventitious roots [6].

Auxins and brassinosteroids (BRs) are two important plant growth regulators that play multiple physiological roles in promoting cell elongation and division, seed germination, fruit development, and plant stress responses [8,9,10,11]. Particularly, extensive reports have been published on auxins and BRs promoting fruit expansion [12,13,14,15,16,17]. Additionally, NAA is an auxin analogue substance that promotes cell division and expansion as a broad-spectrum plant growth regulator [13]. Spraying NAA can improve the fruit size and single-fruit quality of plums, blueberries, guavas, ber, and other fruit crops, promoting higher yield per plant [13,14,15,16]. EBR can improve the yield of wheat, rice, and cucumber, among other crops [9,12,18]. However, there were few reports on the use of growth regulators to increase pumpkin fruit weight; especially the application of NAA and EBR on pumpkin has not been reported yet. Previously, we found that the SNPs alterations in the IAA, BR, and ER pathways may improve fruit enlargement and proposed that auxin and brassinolide may play important roles in giant pumpkin fruit growth [18]. Therefore, this study sought to reveal the effects of applying two growth regulators (NAA and EBR) at different concentrations and at different growth stages on fruit expansion to identify control measures to further improve giant pumpkin-fruit size. The result provides useful information for giant pumpkin growers on how to cultivate larger pumpkins than ever before.

## 2. Results

### 2.1. Effects of NAA and EBR on Giant Pumpkin-Fruit Weight

Great pumpkin single-fruit sprayed with different concentrations of NAA and EBR at 20 days post-anthesis (DPA) showed a higher fruit weight than that of the control (87.6 kg). The single fruit weight in T1 (0.5 mg∙L^−1^ EBR) and T4 (10 mg∙L^−1^ NAA) had no significant (*p* < 0.05) difference relative to the control, while the other 13 treatments showed marked differences. In particular, T10 (20 mg∙L^−1^ NAA + 1.0 mg∙L^−1^ EBR) led to the highest fruit weight, i.e., 112.4 ± 3.8 kg; approximately 1.28-fold higher than the control fruit weight. This was followed by T14 (30 mg∙L^−1^ NAA + 1.0 mg∙L^−1^ EBR) and T11 (20 mg∙L^−1^ NAA + 1.5 mg∙L^−1^ EBR), with values of approximately 107.1 and 106.3 kg, respectively. Thus, spraying pumpkin leaves and fruits with NAA+EBR promoted increased fruit weight, with T10 having the most significant effect (Figure 1).

Furthermore, the effect of the application times of growth regulator combination T10 on pumpkin fruit weight was studied. The fruit weights of multiple application (2–5 times, T16–T19) of 20 mg∙L^−1^ NAA + 1.0 mg∙L^−1^ EBR were significantly higher than that of the control and the one-time application of growth regulator at 20 days after flowering (T10). The fruit weight increased most obviously when NAA+EBR was applied 4 times at 20 d after planting, 0, 10, and 20 DPA, respectively (T18). The single-fruit weight obtained under T18 was approximately 127.2 ± 3.6 kg, indicating an increase of 44.1% and 15.3% compared with that of the control and T10 at 20 DPA, respectively. The effect of NAA and EBR on the increased single-fruit weight was not directly proportional to spraying time. The single-fruit weight obtained under T19 (Applied 20 mg∙L^−1^ NAA + 1.0 mg∙L^−1^ EBR for 5 times) was lower than that of T18, although not significantly (*p* < 0.05) (Figure 2).

### 2.2. Effects of NAA and EBR on the Photosynthetic Characteristics of Giant Pumpkin

The effects of T18 on the photosynthetic characteristics of giant pumpkin at 0 and 20 DPA were assessed (Figure 3). Generally, spraying NAA increased the photosynthesis rate (Pn), stomatal conductance (Gs), and transpiration rate (Tr) but had no significant effect on mesophilic CO_2_ concentration (Ci); however, the effects on Pn, Gs, and Tr were inconsistent. The diurnal variation in the Pn of pumpkin was similar across treatments, which began to increase after 8:00 a.m. and gradually decreased after reaching a peak. The Pn value observed under T18 was notably higher than that recorded for the control. However, at 0 DPA, the maximum value of Pn in the T18 and control groups appeared at 10:00, whereas, at 20 DPA, the maximum value was recorded at 12:00. From 8:00 a.m. to 16:00 p.m., Gs showed a downward trend; however, at 20 DPA, it was higher than that at 0 DPA. Spraying NAA and EBR markedly increased Gs compared with that in the controls before 12:00 at 0 and 20 DPA; however, there was no significant difference in Gs between 14:00 and 16:00 p.m. Tr first increased and then decreased. The peak value of Tr occurred at 14:00 p.m.. Furthermore, at 20 DPA, Tr was substantially higher than at 0 DPA, and T18 led to higher Tr than those recorded for the controls. Pn, Gs, and Tr at 20 DPA were higher than those at 0 DPA, indicating that more assimilates must be produced and more water must be supplied during the rapid fruit-expansion period, and the T18 treatment obviously increased the photosynthesis rate.

### 2.3. Effects of NAA and EBR on Sugar-Metabolism-Related Gene Expression and Enzyme Activities in Giant Pumpkin Leaves

The expression levels of galactinol synthase *(CmGolS1*), raffinose synthetase (*CmRS*), and stachyose synthetase *(CmSTS*) were consistent with changes in enzyme activity during the growth and development of giant pumpkin leaves. The expression of *CmGolS1* was higher in the early stages of fruit development and lower after 50 d. In turn, *CmRS* expression was maintained at a relatively stable level during fruit development, while *CmSTS* expression increased after the fruit entered the rapid growth period and gradually decreased after 50 d (Figure 4A–C). The NAA+EBR combined treatment enhanced the expression of *CmGolS1*, *CmRS*, and *CmSTS* and the corresponding enzyme activity of GolS, RS, and STS of Giant Pumpkin. (Figure 4D–F). Overall, spraying growth regulators promoted stachyose synthesis in pumpkin leaves and can be used to increase the production of photoassimilate for rapid fruit growth.

### 2.4. Effect of NAA and EBR on Soluble Sugar Content in Giant Pumpkin Leaves

Stachyose, raffinose, sucrose, glucose, and fructose contents in pumpkin leaves gradually decreased with fruit growth and development. However, the time at which sugar content began to decrease was different (Figure 5). Stachyose and raffinose contents remained high during the early stages of fruit development, but they were notably reduced after 40 DPA, particularly raffinose, which declined by approximately 42.3%, whereas stachyose content decreased by 19.6% at 60 DPA. Sucrose content decreased gradually at 30 DPA and increased slightly in the later stages of fruit development. Glucose and fructose contents began to drop from 0 DPA, reaching a minimum at 20 DPA. The application of 20 mg∙L^−1^ NAA + 1.0 mg∙L^−1^ EBR four times increased the leaf stachyose content, especially in the late stage of fruit development. Although stachyose decreased at 60 DPA, a high stachyose content was maintained in the NAA+EBR-treated leaves and was only reduced by 15.1%. In contrast, stachyose content in control leaves decreased by 24.2%.

### 2.5. Effects of NAA and EBR on Soluble Sugar Content in the Phloem Sap of Giant Pumpkin Peduncles

Stachyose and sucrose contents in the phloem sap of the pumpkin peduncle tended to be the same in controls and NAA+EBR treatment groups. Overall, stachyose and sucrose contents increased with fruit growth and development and were higher with NAA+EBR than those of the controls, with increases of approximately 1.3–3.1 and 0.8–2.5 mg∙g^−1^, respectively (Figure 6A,C). Raffinose levels increased and then decreased during fruit development, reaching a maximum value at 20 DPA and a minimum value at 60 DPA. However, spraying with NAA+EBR had no significant effect (Figure 6B). Though glucose content was consistent with the change in raffinose, NAA and EBR increased glucose content during the first 30 d of fruit development (Figure 6D). Fructose content generally showed a downward trend and was almost zero at 30 DPA. However, at 10 DPA, fructose content in the phloem sap sprayed with NAA and EBR was higher than that of the controls (Figure 6E). The use of two plant growth regulators on fruit and leaves increased the concentrations of stachyose and sucrose throughout pumpkin fruit growth, in addition to increasing glucose and fructose contents at 0 and 20 DPA, but had little effect on raffinose in the phloem sap of the giant pumpkin peduncle. Thus, the NAA+EBR combined treatment increased photoassimilate transport efficiency.

### 2.6. Effects of NAA and EBR on Alkaline α-Galactosidase in Giant Pumpkin Fruits

Under normal conditions, the expression of alkaline α-galactosidase *CmAGA1* and *CmAGA2* in pumpkin fruits showed almost no change during fruit development; however, AGA activity increased and then decreased (Figure 7), and the maximum was observed at the fruit rapid-expansion stage (20 DPA). The expression levels of *CmAGA1* and *CmAGA2* and the activity of AGA increased in pumpkin fruit upon the exogenous application of NAA+EBR, indicating an enhancement of assimilate metabolism in the giant pumpkin fruit.

### 2.7. Effects of NAA and EBR on Soluble Sugar Content in Giant Pumpkin Fruit

Stachyose content in pumpkin fruit reached a maximum on day 20 of pumpkin development and then decreased rapidly. In turn, raffinose content decreased rapidly after anthesis, with the minimum content obtained at 20 DPA, followed by a slight increase. NAA+EBR did not affect the contents of these two raffinose family oligosaccharides (RFOs). Changes in sucrose and fructose contents during fruit development were consistent with those of stachyose. Meanwhile, glucose content reached a maximum value at 10 DPA and then gradually decreased, aligning with raffinose. Lastly, fruit sucrose, glucose, and fructose contents after NAA+EBR treatment were higher than those in the control treatment (Figure 8). Thus, spraying NAA+EBR on giant pumpkin leaves and fruit was beneficial for the unloading of stachyose and raffinose in the fruit and providing energy for fruit growth.

## 3. Discussion

To reach maturity, fruit growth depends on photoassimilate from source organs to enhance sink strength by supporting cell division and growth, ultimately promoting the transfer of photoassimilate to the growing fruits [19]. NAA is a synthetic auxin with the similar physiological function as IAA, and owing to its lower cost and greater stability, it is more suitable for application in crop production. The effects of NAA on horticultural crops have been widely studied. For example, Stern et al., 2009, applied NAA to plums and found increased single-fruit quality, size, and total soluble solid content [14]. Gill and Bal, 2009, reported similar results when NAA was sprayed on ber leaves [13]. Furthermore, Benedetto et al., 2015, found that exogenous auxin promoted an increase in net assimilation and Pn in *Epipremum aureum* [10]. In this study, the Pn of pumpkins treated with NAA increased, indicating an improved carbon uptake capacity. Effectively improving leaf photosynthetic performance by the exogenous application of NAA may be related to the increase of leaf area and chlorophyll content and the improvement of stomatal and non-stomatal restrictions, as well as the enhancement of photosynthetic system II [20,21]. After fruit setting, most of the assimilates synthesized via photosynthesis are used for fruit growth, except for physiological activities such as growth and respiration. Auxins improve the photosynthetic capacity of leaves and regulate the loading, transport, and distribution of photoassimilate, thereby improving the seed setting rate and increasing grain weight [22,23]. Singh et al., 2017, found that the total soluble solids, reducing sugars, ascorbic acid, and total sugar contents of guava fruit treated with NAA increased [15]. Pumpkin uses RFOs as the main form of photoassimilate. The key enzymes required for RFO synthesis are GloS, RS, and STS, encoded by *CmGolS1*, *CmRS*, and *CmSTS*, respectively [1,4,24,25]. In this study, plant growth regulators notably increased the expression and activity of GolS, RS, and STS, which promoted stachyose synthesis and enhanced the ability of leaves to engage in the loading of assimilates. In plants in which photosynthetic products are transported as sucrose, exogenous auxin can promote sucrose content and the expression level of sucrose transporter-encoding genes by increasing the grain filling rate and the unloading of phloem sucrose into the fruit [26,27]. In the phloem sap of giant pumpkin peduncles, spraying NAA can increase stachyose and sucrose contents, indicating that treatment with this growth regulator improves assimilate transport efficiency. AGA, which is a key enzyme involved in stachyose unloading [1,4], is encoded by three pumpkin genes, among which *CmAGA1* and *CmAGA2* are important for stachyose catabolism. In this study, we found that AGA activity and the expression of *CmAGA1* and *CmAGA2* were markedly increased in NAA-treated leaves, indicating an enhancement of stachyose metabolism in fruit and an increase in the substances needed for fruit growth.

BRs mediate many physiological processes in plants, including increasing leaf area, regulating photosynthesis, promoting carbohydrate metabolism, increasing plant biomass, and increasing grain size and yield [8,11,17,28,29,30]. EBR are some of the most widely used BR compounds in plants. Spraying exogenous EBR on pumpkin leaves and fruits effectively increased fruit weight, with the maximum obtained by EBR treatment at 1.0 mg∙L^−1^. This result is similar to others recently reported [12,18]. BRs participate in the regulation of plant photosynthetic characteristics and are known to promote Pn [8,31]. Applying EBR to cucumbers can effectively increase seedling growth and yield, which is related to the improvement of the CO_2_ assimilation level [9,32,33]. The effects of ETR treatment on photosynthetic capacity might be mediated by the increased chlorophyll content and regulation of photosynthetic enzymatic activity [8,33,34,35]. In this study, EBR significantly improved the Pn, Gs, and Tr of giant pumpkin, indicating that EBR enhanced photosynthetic performance, increasing leaf soluble sugar content and sink strength. Sugar unloading from fruit is a key step for the long-distance transport of assimilates from leaves (source organs) to fruits (sink organs). Sugar unloading activity depends on sink strength. For plants that transport sucrose as the main photoassimilate, the activities of the sucrose-metabolism-related enzymes, invertases (INVs), and sucrose synthases (SuSyn) are key indicators of sink strength [28,36]. BRs reportedly increase sugar content, (i.e., sucrose, fructose, or total soluble sugar) and SuSyn and INVs activities in wheat [17] and grape [28] and upregulate mRNA levels of sucrose-metabolism- and transport-related genes in grapes [28]. Previous studies have shown that sink activity is an important factor in determining fruit size, and giant pumpkins have an enhanced capacity to assimilate carbon [5,18]. Pumpkin is a raffinose-transport plant whose sink activity is closely related to the activity and expression of AGA [4,25]. In this study, EBR increased the leaf stachyose content, improved the gene expression and activity of AGA in fruit, promoted more organic matter entry into the fruit and ultimately increased fruit weight.

Auxin and BRs have cooperative effects and many similar physiological functions, such as synergistic involvement in the formation of lateral roots. In addition, it was also reported that BRs regulated auxin polar transport [37]. NAA or EBR applied separately increased the weight of giant pumpkins, and the combination treatment further increased fruit weight markedly. Plant hormones have dual characteristics: they promote growth at low concentrations but inhibit it at high concentrations [13,38]. Additionally, An et al., 2019, reported that exogenous NAA regulates strawberry growth through multiple metabolic pathways, and different NAA application times also affect these regulatory effects [39]. The concentrations of NAA and EBR and the number of applications were not directly proportional to the increase in pumpkin fruit weight. Giant pumpkin growth was not inhibited by the highest concentration (30 mg∙L^−1^) of exogenous NAA, which led to a higher fruit weight than that of the control, although notably lower than that observed when 20 mg.L^−1^ exogenous NAA was applied. Pumpkin fruit weight with five exogenous applications of NAA+EBR was significantly lower than that observed with four applications of NAA+EBR. This result may be due to insufficient concentrations of exogenous NAA and EBR to inhibit pumpkin growth. Nonetheless, the effects of high concentrations of NAA and EBR on pumpkin fruit weight decreased. Because of the high temperature in the pumpkin cultivation area during the summer, cultivation in this season is not convenient, as heat markedly restricts fruit growth. This may be an important reason for the evident difference in fruit weight relative to that reported in the world records. Fruit weight significantly increased by applying NAA and EBR; however, this increase was limited. Further research is required to determine whether the fruit weight of giant pumpkins can be adjusted by increasing the nutrient area using growth regulators or improving horticultural cultivation facilities or practices to reduce summer temperature and extend the fruit growth period in this cultivation region. Although this study did not analyze the residues of NAA and EBR in pumpkins, a large number of previous studies showed that NAA dissipated rapidly in plants, and the residues of NAA were lower than the safety limit of pesticides in agricultural products and food (maximum residue limit, MRL), and EBR residues could not be detected in the plants in which EBR content was widely used [40,41,42,43,44].

Taken together, exogenous NAA+EBR increased pumpkin fruit weight via an all-around coordinated action involving the source (leaf), transport (peduncle phloem sap), and the sink (fruit). Application of NAA+EBR can improve the photosynthetic capacity of leaves (sources), promote the synthesis of assimilates and water supply, enhance the transport efficiency of assimilates from the peduncle phloem sap to the growing fruit, and also improve sink strength, which was conducive to the unloading of assimilates, thus providing carbon skeletons and energy for fruit growth, ultimately promoting the increase in pumpkin fruit weight. According to the results, the recommended combination of growth regulators for pumpkin production was 20 mg∙L^−1^ NAA + 1.0 mg∙L^−1^ EBR, and the optimal periods of application were spraying on the 20 days after planting, 0, 10, and 20 DPA.

## 4. Materials and Methods

### 4.1. Plant Materials and Cultivation

The Atlantic giant pumpkin, which belongs to the Indian variety, was purchased from the Sustainable Seed Company (Covelo, CA, USA). Pumpkin seeds were sown on 1 February 2021, and, at the three-leaf stage, seedlings were transplanted into the plastic greenhouse of Lehou Manor Agricultural Ecological Park, in Shatou Town, Yangzhou City, on 10 March. Pumpkin seedlings were planted in 50 cm-deep holes approximately 1–1.5 m in diameter; 10 kg of compound fertilizer (N: P_2_O_5_: K_2_O = 15:15:15), 10 kg of CaSO_4_ fertilizer, 10 kg of rape seed leaves, and 20 kg of cow dung were applied to each planting hole. The row spacing was 5 m × 2 m. The main vine of the plants retained 40 knots, and the secondary vine retained 3–4 knots. Each plant retained only one fruit (i.e., the second fruit on the main stem).

### 4.2. Experimental Design

#### 4.2.1. Design of NAA and EBR Combination Treatments

During the rapid fruit expansion period (20 DPA), giant pumpkin leaves (both sides) and fruits were sprayed with different concentrations of NAA and EBR until dripping, and distilled water (Tween-20 with a volume fraction of 0.1%) was used as the control. Each treatment was repeated 3 times, and each repetition included 20 plants. Specific concentrations of NAA and EBR are listed in Table 1.

#### 4.2.2. NAA and EBR Period and Frequency of Application

To further elucidate the beneficial effect of NAA and EBR on giant pumpkin fruit-weight, the combination treatment producing the largest single-fruit weight in Table 1 was selected and then applied at different stages with different times during pumpkin growth. The specific design for NAA and EBR application periods and times are listed in Table 2.

### 4.3. Sample Collection

After treatment, samples were collected at 10-day intervals, and the leaves of the fruit-setting node and fruits were collected; this was repeated three times for every five leaves or fruits. The phloem sap of the pumpkin peduncle was collected as previously described [4,6,45]. Peduncles were removed from the fruit, and the phloem sap was collected from the side near the stem using a capillary tube. The sap collected in the first 2 s was discarded to reduce the cell and xylem sap contamination. All samples were frozen in liquid nitrogen immediately after collection and stored at −80 °C.

### 4.4. Measurement Methods

#### 4.4.1. Determination of Single-Fruit Weight

Pumpkin fruits were harvested when growth stopped, and single-fruit weight was measured using an electronic scale. 60 pumpkin fruits were measured for each treatment and the average value was displayed.

#### 4.4.2. Determination of Photosynthetic Characteristics

Net photosynthetic rate (Pn), mesophyll CO_2_ concentration (Ci), stomatal conductance (Gs), and transpiration rate (Tr) of the leaves located on the fruiting-node were measured using a portable photosynthesis measurement system Li-6400XT (LI-COR, Lincoln, NE, USA) from 8:00 to 16:00 h at 0 and 20 DPA. The measurements were made five times at 2 h intervals. Five pumpkin plants from each cultivar were measured at each measuring time.

#### 4.4.3. Determination of the Gene Expression of Sugar-Metabolism-Related Enzyme Encoding Genes

Total RNA was extracted from pumpkin leaves and fruits using an RNAiso Plus kit (Takara Bio Inc., Shiga, Japan), according to the manufacturer’s instructions. Reverse transcription was performed using EasyScript^®^ One-Step gDNA Removal, and cDNA Synthesis Super Mix (TransGen Biotech Co. Ltd., Peking, China) was performed for cDNA synthesis according to the manufacturer’s instructions. iTaq Universal SYBR^®^ Green Supermix (Bio-Rad, Hercules, CA, USA) was used to determine the expression of related genes, according to the manufacturer’s instructions. The reaction system comprised reaction mix 12.5 μL, cDNA 1.0 μL, 0.5 μL for the forward primer, 0.5 μL for the reverse primer, and 5.5 μL double distilled water (ddH_2_O). Polymerase chain reaction (PCR) amplification was performed using a CFX96 Touch Real-Time PCR Detection System (Bio-Rad, Hercules, CA, USA). The cycling program was as follows: 94 °C for 2 min; 35 cycles of 94 °C for 10 s, 60 °C for 5 s; and a slow temperature increase from 65 to 95 °C at a rate of 0.5 °C ∙min^−1^. The expression level of each gene was normalized to that of the pumpkin internal reference gene, *EF-1a*, and calculated using the 2^−ΔΔCT^ method. Primers used in this study are listed in Appendix A.

#### 4.4.4. Determination of Sugar-Metabolism-Related Enzyme Activities

To detect GolS, RS, and STS, 0.5 g of the frozen ground sample was homogenized with 2 mL 50 mmol∙L^−1^ HEPES-NaOH buffer (pH 7.0, 2 mmol∙L^−1^ DTT, 10 mg∙mL^−1^ PVP) and centrifuged (18,000× *g*, 30 min). Supernatants were retained and dialyzed overnight with a buffer containing 25 mmol∙L^−1^ HEPES-NaOH buffer and 1 mmol∙L^−1^ DTT, and the enzyme solution was collected.

Activities of GolS, RS, and STS were measured according to the methods of Li et al., 2011 [46], and Wang et al., 2012 [47]. The 200 μL reaction system contained 100 mmol∙L^−1^ HEPES-NaOH buffer (pH 7.0), 20 mmol∙L^−1^ β-mercaptoethanol, 5 mmol∙L^−1^ MgCl_2_, and 4 mmol∙L^−1^ DTT; and different enzymes were added to different reaction substrates. Specifically, the reaction system for determining GolS activity also contained 40 mmol∙L^−1^ sucrose, 20 mmol∙L^−1^ inositol, and 5 mmol∙L^−1^ UDP galactose; in turn, the system for determining RS activity contained 20 mmol∙L^−1^ inositol galactoside; the system for determining STS activity contained 40 mmol∙L^−1^ raffinose and 20 mmol∙L^−1^ galactoside. Subsequently, 50 μL of the enzyme solution was added to a water bath at 30 °C for 3 h and boiled for 5 min. After cooling, high-performance liquid chromatography (HPLC)-Agilent Technologies 1260 Infinity (Agilent Technologies, Santa Clara, CA, USA) was used for the detection of enzymatic activities, which were expressed as the amounts of galactoside, raffinose, and stachyose produced for GolS, RS, and STS, respectively.

AGA was extracted as previously described by Miao et al., 2007 [48], and Wang et al., 2016 [49]. Frozen samples (0.25 g) were added to 1 mL of the extraction solution (50 mmol∙L^−1^ HEPES-NaOH buffer with 2 mmol ∙L^−1^ MgCl_2_, 1 mmol∙L^−1^ EDTA, and 1 mmol ∙L^−1^ DDT, pH7.4) for 15 min at 25 °C and centrifuged at 18,000× *g*∙min^−1^ for 30 min. Then, 5% (*w*/*v*) polyethylene glycol (PEG6000) was added to the supernatant, prior to centrifugation for 30 min. This step was repeated with 50% (*w*/*v*) PEG6000. The precipitate was redissolved in 25 mmol∙L^−1^ HEPES NaOH buffer with 1 mmol∙L^−1^ DTT and dialyzed overnight. The assay was performed using a 100 μL reaction system containing 100 mmol∙L^−1^ HEPES NaOH buffer (pH 7.4) and 16 mmol∙L^−1^ stachyose; 30 μL of the extracted enzyme was also added. After 15 min, 100 μL of 5% (*w*/*v*) Na_2_CO_3_ was added, and the mixture was boiled for 5 min to terminate the reaction. The amount of stachyose consumed was measured using HPLC to determine enzyme activity.

#### 4.4.5. Extraction and Determination of Soluble Sugar

Soluble sugars were extracted and determined in pumpkin tissues according to Miao et al., 2007 [48], and Zhang et al., 2020 [24]. After 0.3 g frozen tissue was fully ground, extraction was performed using 10 mL of 80% ethanol for 10 min, and the resulting extract was centrifuged at 6000× *g*∙min^−1^ for 15 min. The supernatant was then placed in a rotary evaporator, evaporated under reduced pressure at 40 °C, dissolved in ddH_2_O and chloroform (2.5 mL), shaken, and centrifuged to remove the organic phase. The pH of the aqueous phase was adjusted to 7.0 using NaOH (0.1 mol∙L^−1^ NaOH) and evaporated to dryness under reduced pressure. Finally, the solution was dissolved in 1 mL ddH_2_O and passed through a 0.45 μm needle filter. Using the same chromatographic conditions as the standard, the stachyose, raffinose, sucrose, glucose, and fructose contents were calculated using the external standard method. For HPLC, an Agilent Hi-Plex Ca (Duo) (Agilent Technologies, Santa Clara, CA, USA) column (300 mm × 6.5 mm) was used as the chromatographic column; ultrapure water was used as the mobile phase, and the column temperature was 80 °C; the flow rate was 0.5 mL∙min^−1^; and the injection volume was 5 μL. Retention time and peak area were used to qualitatively and quantitatively analyze soluble sugar contents.

### 4.5. Statistical Analysis

Data were sorted using Microsoft Excel 2016 (Microsoft Corporation, Redmond, WA, USA). Meanwhile, SPSS Statistics 24.0 (IBM, Chicago, IL, USA) was used to determine significant differences among treatments. GraphPad Prism v9.0 (GraphPad Software Inc., San Diego, CA, USA) was used to visualize and generate plots.

## Figures and Tables

**Figure 1 ijms-23-13157-f001:**
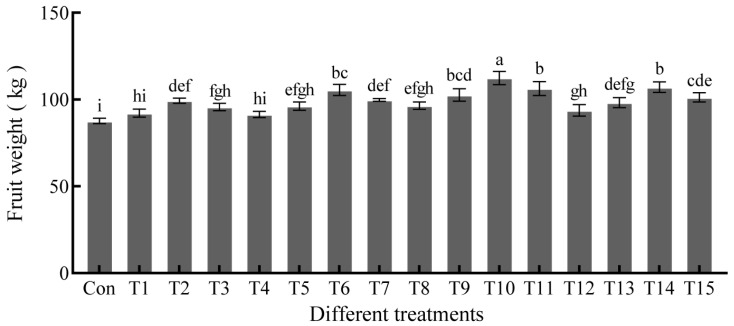
Effects of different concentrations of NAA and EBR on the fruit weight of giant pumpkin. Different lowercase letters on the bars indicate significant difference (*p*-value < 0.05). Con: control; giant pumpkin leaves and fruits sprayed distilled water (Tween-20 with a volume fraction of 0.1%), and “Con” represented the same meaning as the following. T: growth regulator treatment, giant pumpkin leaves and fruits sprayed with NAA and EBR, and the details of T1–14 combination are listed in Section 4.

**Figure 2 ijms-23-13157-f002:**
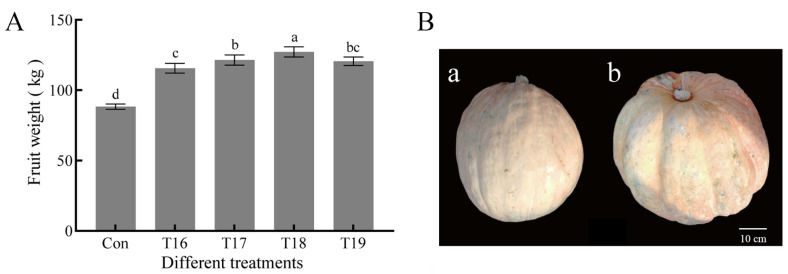
Effect of application period and times of NAA and EBR treatment on giant pumpkin-fruit weight. (**A**) Fruit weight. Different lowercase letters on the bars indicate significant difference (*p*-value < 0.05). The details of the T16–T18 combination are listed in Section 4. (**B**) Fruit morphology. a, Con, b, T18, giant pumpkin leaves and fruits sprayed 20 mg∙L^−1^ NAA + 1.0 mg∙L^−1^ EBR at 20 d after planting, 0 DPA, 10 DPA, and 20 DPA, respectively.

**Figure 3 ijms-23-13157-f003:**
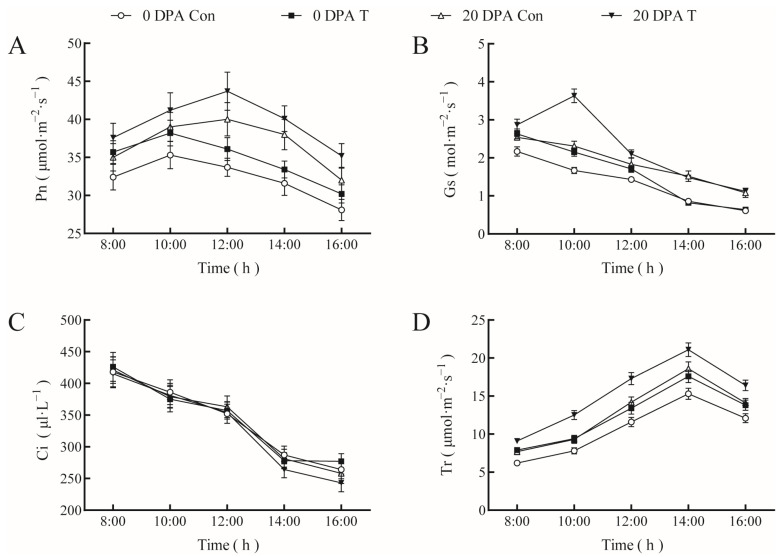
Effects of NAA and EBR on the photosynthetic characteristics of giant pumpkin leaves. (**A**), Pn, (**B**), Gs, (**C**), Ci, (**D**), Tr. Con: control; T: giant pumpkin leaves and fruits sprayed with 20 mg∙L^−1^ NAA + 1.0 mg∙L^−1^ EBR at 20 d after planting, 0 DPA, 10 DPA, and 20 DPA. The “Con” and “T” represent the same meaning as the following.

**Figure 4 ijms-23-13157-f004:**
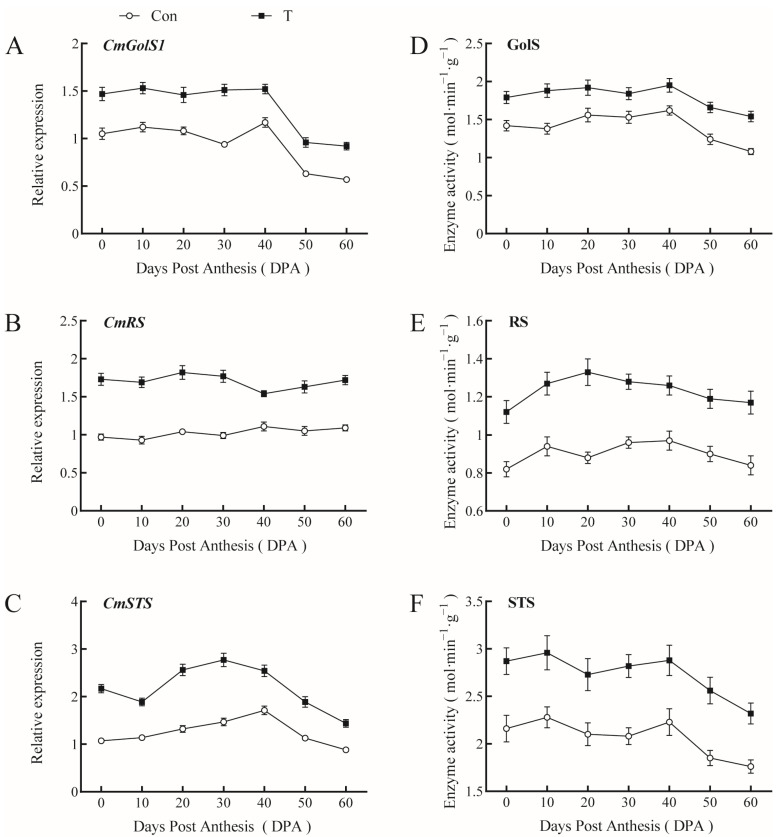
Effects of NAA and EBR on sugar-metabolism-related gene expression and enzyme activities in giant pumpkin leaves. (**A**–**C**) Relative expression levels of *CmGolS1*, *Cm**RS*, and *CmSTS*; (**D**–**F**) Activities of GolS, RS, and STS.

**Figure 5 ijms-23-13157-f005:**
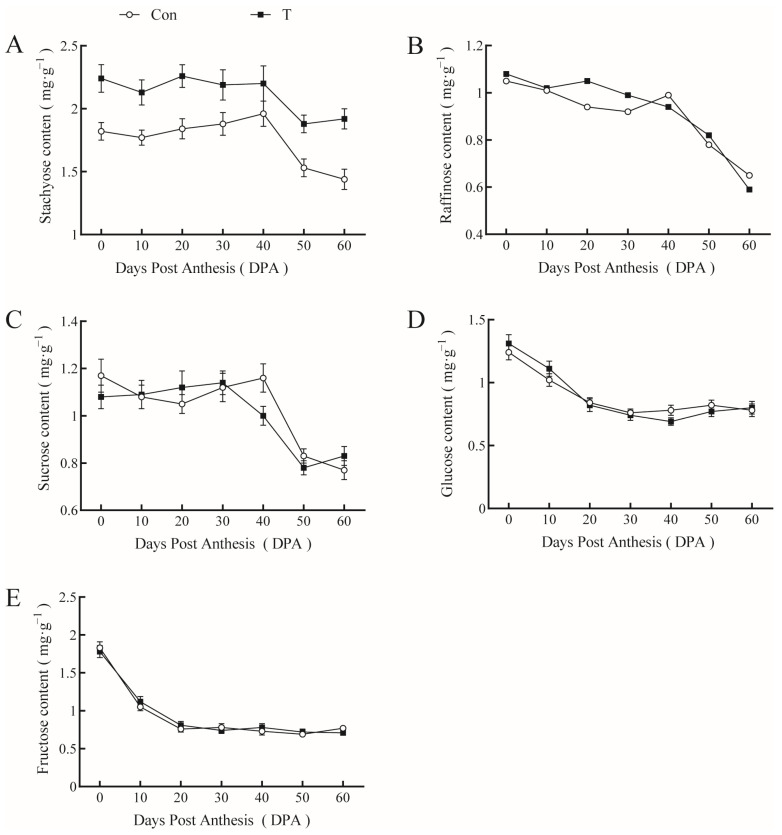
Effect of NAA and EBR on soluble sugar content in giant pumpkin leaves. (**A**) Stachyose, (**B**) raffinose, (**C**) sucrose, (**D**) glucose, (**E**) fructose.

**Figure 6 ijms-23-13157-f006:**
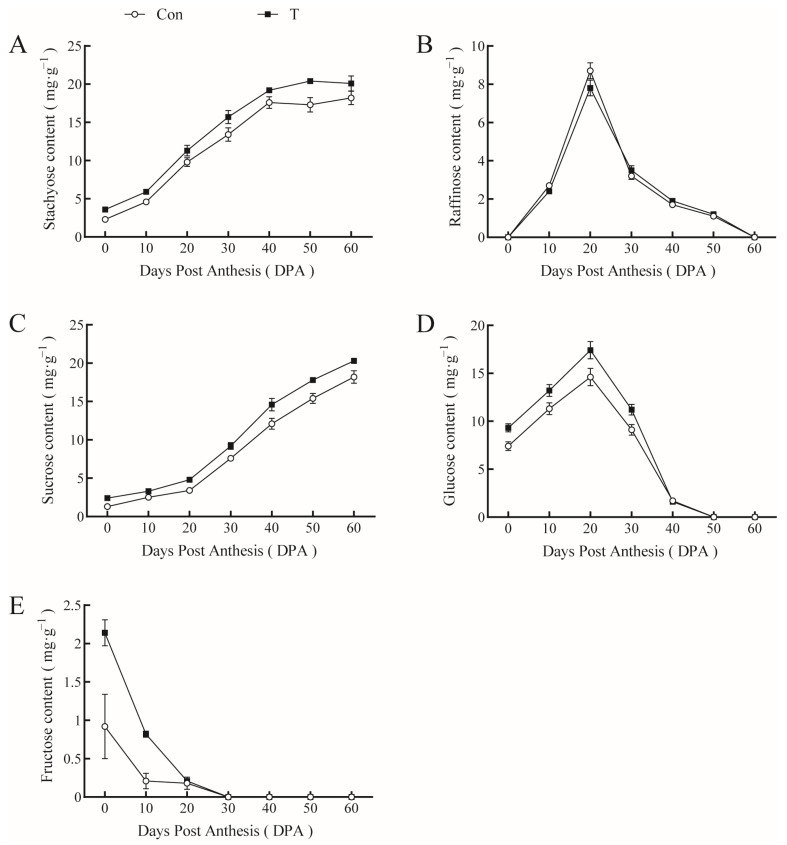
Effect of NAA and EBR on soluble sugar content in the phloem sap of giant pumpkin. (**A**) Stachyose, (**B**) raffinose, (**C**) sucrose, (**D**) glucose, (**E**) fructose.

**Figure 7 ijms-23-13157-f007:**
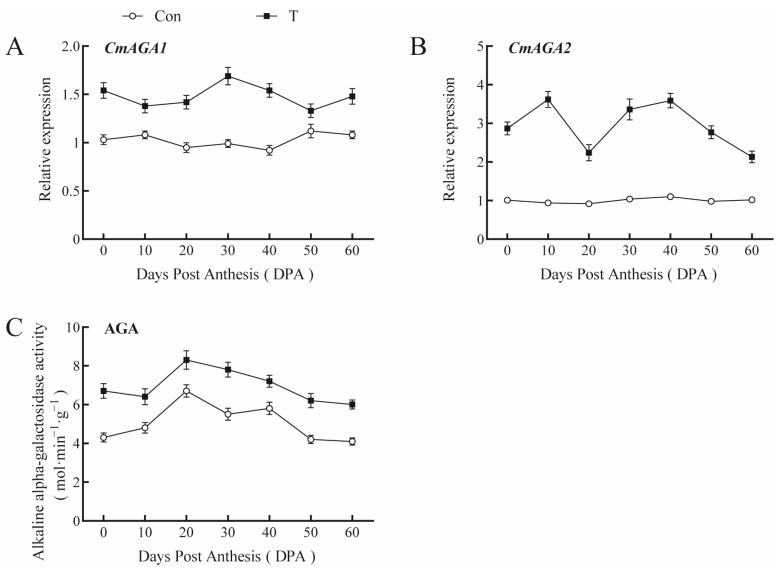
Effect of NAA and EBR on gene expression and activity of alkaline α-galactosidase (AGA) in giant pumpkin fruits. (**A**,**B**), Relative expression levels of *CmAGA1* and *CmAGA2*; (**C**) activity of AGA.

**Figure 8 ijms-23-13157-f008:**
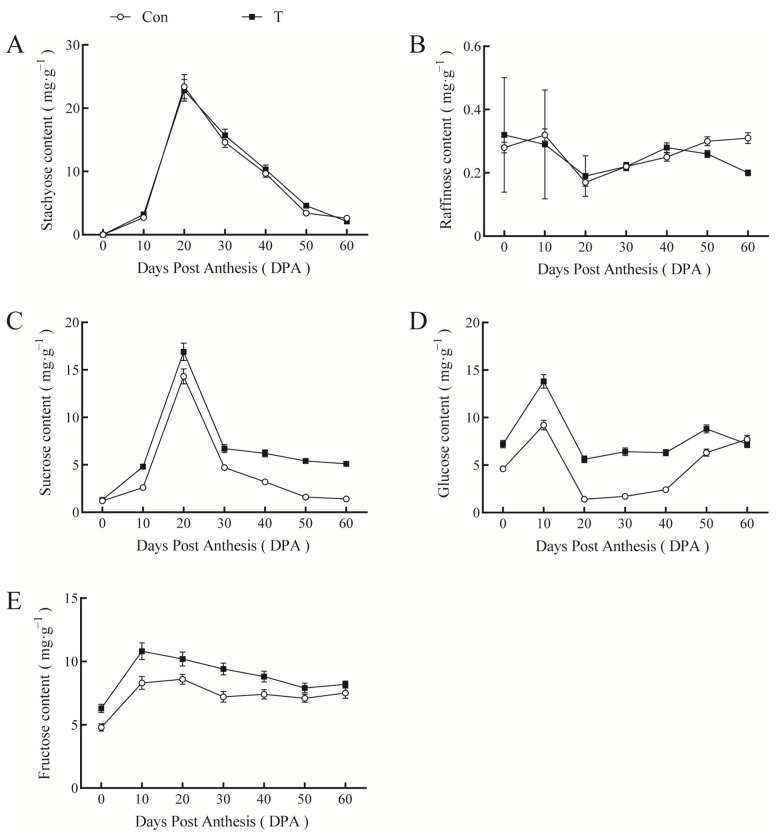
Effect of NAA and EBR on soluble sugar content in giant pumpkin fruit. (**A**) Stachyose, (**B**) raffinose, (**C**) sucrose, (**D**) glucose, (**E**) fructose.

**Table 1 ijms-23-13157-t001:** Different combinations of NAA and EBR.

Treatments	NAA mg∙L^−1^	EBR mg∙L^−1^	Treatments	NAA mg∙L^−1^	EBR mg∙L^−1^
Control	0	0	T8	20	0
T1	0	0.5	T9	20	0.5
T2	0	1.0	T10	20	1.0
T3	0	1.5	T11	20	1.5
T4	10	0	T12	30	0
T5	10	0.5	T13	30	0.5
T6	10	1.0	T14	30	1.0
T7	10	1.5	T15	30	1.5

**Table 2 ijms-23-13157-t002:** NAA and EBR application period and frequency.

Treatments	Times	Application Period				
		20 d after Planting	0 DPA	10 DPA	20 DPA	30 DPA
Control	0	—	—	—	—	—
T16	2	—	S	—	S	—
T17	3	—	S	S	S	—
T18	4	S	S	S	S	—
T19	5	S	S	S	S	S

Note: “S”, giant pumpkin leaves and fruits sprayed NAA+EBR; “—”, NAA+BER was not applied.

## Data Availability

Not applicable.

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
