# Peer review of "Effects of Exogenous α-Naphthaleneacetic Acid and 24-Epibrassinolide on Fruit Size and Assimilate Metabolism-Related Sugars and Enzyme Activities in Giant Pumpkin"

_ijms, 2022, doi:10.3390/ijms232113157_

Round 1
Reviewer 1 Report
The manuscript focuses on the desired alteration of physical parameters in pumpkin varieties and carries and overall practical merit. Since the authors focus on the modulation of size and weight of the fruit, the manuscript might benefit from the addition of relative photographs corresponding to the specific treatments.
Apart from that, the manuscript should be double-checked for formatting flaws like in L17 and 31 where the alpha sign is missing, L45-6 with an unnecessary line space, etc.
Author Response
Response to Reviewer 1 Comments
Point 1: The manuscript focuses on the desired alteration of physical parameters in pumpkin varieties and carries and overall practical merit. Since the authors focus on the modulation of size and weight of the fruit, the manuscript might benefit from the addition of relative photographs corresponding to the specific treatments.
Response 1: Following the reviewer’s advice, we added the fruit morphology in Fig.2 B of the Results section “2.1. Effects of NAA and EBR on giant pumpkin-fruit weight”.
Point 2: Apart from that, the manuscript should be double-checked for formatting flaws like in L17 and 31 where the alpha sign is missing, L45-6 with an unnecessary line space, etc.
Response 2: Following the reviewer’s advice, we used the spell checker to examine the text, and added the α in line 17 and 31, and the line space in L45-6 was deleted.

Author Response
Response to Reviewer 2 Comments
Point: The manuscript “Effects of Exogenous α-Naphthaleneacetic Acid and 24-Epibrassinolide on Fruit Weight and Assimilate Metabolism-Related Sugars and Enzyme Activities in Giant Pumpkin” is an original work and provides useful information, is sufficiently complete and comprehensive, clearly written and executed based on appropriate, reliable and updated technologies. This work is really interesting and provide information about using α-Naphthaleneacetic Acid and 24-Epibrassinolide to improve the fruit quality of horticultural crops. wild berries to be used as ingredients in many products, as you mention. It was a hard work to collect fruits during different periods, prepare samples, analyze them and finally process data. I would only suggest to indicate the novelty of the text and results.
Response: Following the reviewer’s advice, the sentences were added in the third paragraph of Introduction section:
Line 66-68:However, there were few reports on the use of growth regulators to increase pumpkin fruit weight, especially the application of NAA and EBR on pumpkin has not been reported yet.
In addition, the following sentence was added at the end of the introduction:
Line 74-75: The result provides useful information for giant pumpkin growers how to culture larger pumpkins than ever before.

Reviewer 3 Report
The manuscript entitled “Effects of Exogenous α-Naphthaleneacetic Acid and 24 Epibrassinolide on Fruit Size and Assimilate Metabolism-Related Sugars and Enzyme Activities in Giant Pumpkin” comprehensively evaluates the response of spraying different concentrations-naphthaleneacetic acid (NAA) and 24-epibrassinolide (EBR) on the giant pumpkin’s leaves and fruit at different frequencies and growth stages determine their effects and the mechanism responsible for fruit size increase. Therefore, the study is of great importance and holds the potential to be considered for publication in International Journal of Molecular Sciences. However, there are some issues that must be addressed for its suitability.
It is not clear in the manuscript how different pumpkin fruit at different stages responded to application of growth regulators, to me it seems as if the application was only on one stage of development, please clarify.
Abstract: The abstract presented is informative with the clear aim of the study
Keywords: They seem adequate but not written in any order
1. Introduction
Line 36- Instead of the peel, I suggest that you put ’its’ peel…..
In your introduction, I suggest you highlight the importance of both α-naphthaleneacetic acid (NAA) and 24-Epibrassinolide (EBR) in regulating growth and yield, as they are the major treatments used in the present study. If these treatments were once applied to pumpkins individually, what were the important findings, you can further mention that the combination of the two growth regulators has never been tested on giant pumpkins therefore this is what your study is trying to address. This will ultimately indicate if this study has low or high novelty.
In line 66-67 you highlighted that “Previously, we proposed that auxin and brassinolide may play important roles in giant pumpkin”, was the study on pumpkins, and if so, briefly mention the affected parameters?
2. Materials and methods
Plant materials and cultivation
How did you arrive at the amount of fertilizer (organic and inorganic) to apply to the pumpkin plants, are they standard fertilizers? If too much was applied, how sure you are that the effects observed were a result of α-naphthaleneacetic acid (NAA) and 24-Epibrassinolide (EBR)?
There is T10 in both tables 1 and 2, how is that possible?
It is hard to understand table 2, please make it easy to understand the information presented in it.
In line 387, fix the °C symbol
In line 396, there must be a space between with and 50%
3. Results
Line 76- I don’t think it’s appropriate to start a new line with ‘except’ after a full stop
Line 85- ‘’ fruit Fw’’ there is something wrong here, please fix, same in line 88…check throughout
Line 160- remove a full stop after %
As from section 2.2. why were the effects only referred to T18
Overall, even though you presented enough data, it is however not easy for the reader to follow your results, try to be more precise and mention the most important results. In addition, avoid discussing your results in this section, only describe them.
4. Discussion
Line 273- …it also promoted….
Overall, the discussion is comprehensive

Author Response
Response to Reviewer 3 Comments
Point 1: The manuscript entitled “Effects of Exogenous α-Naphthaleneacetic Acid and 24 Epibrassinolide on Fruit Size and Assimilate Metabolism-Related Sugars and Enzyme Activities in Giant Pumpkin” comprehensively evaluates the response of spraying different concentrations-naphthaleneacetic acid (NAA) and 24-epibrassinolide (EBR) on the giant pumpkin’s leaves and fruit at different frequencies and growth stages determine their effects and the mechanism responsible for fruit size increase. Therefore, the study is of great importance and holds the potential to be considered for publication in International Journal of Molecular Sciences. However, there are some issues that must be addressed for its suitability.
It is not clear in the manuscript how different pumpkin fruit at different stages responded to application of growth regulators, to me it seems as if the application was only on one stage of development, please clarify.
Response 1: At first, we applied the growth regulators at the rapid fruit expansion stage, as mentioned in Line 342-343: “During the rapid fruit expansion period 20 DPA, giant pumpkin leaves (both sides) and fruits were sprayed with different concentrations of NAA and EBR until dripping”
And then, we selected the best treatment and conduct another experiment to explore the optimal period and time of the treatment, as mentioned in Line 350-353: “To further elucidate the beneficial effect of NAA and EBR on giant pumpkin fruit-weight, the combination treatment producing the largest single-fruit weight in Table 1 was selected, and then applied at different stages with different times during pumpkin growth. “
Keywords
Point 2: Keywords: They seem adequate but not written in any order.
Response 2: The keywords were modified as follows: giant pumpkin; É‘-Naphthaleneacetic acid; 24-epibrassinolide; source capacity; photoassimilate; sugar transport; sink strength; fruit size
- Introduction
Point 3: Line 36- Instead of the peel, I suggest that you put ’its’ peel…..
Response 3: Line 36, we added ’its’ before “peel”.
Point 4: In your introduction, I suggest you highlight the importance of both α-naphthaleneacetic acid (NAA) and 24-Epibrassinolide (EBR) in regulating growth and yield, as they are the major treatments used in the present study. If these treatments were once applied to pumpkins individually, what were the important findings, you can further mention that the combination of the two growth regulators has never been tested on giant pumpkins therefore this is what your study is trying to address. This will ultimately indicate if this study has low or high novelty.
Response 4: Following the reviewer’s advice, the paragraph was rewritten in the third paragraph of Introduction section:
Line 66-68:However, there were few reports on the use of growth regulators to increase pumpkin fruit weight, especially the application of NAA and EBR on pumpkin has not been reported yet.
Point 5: In line 66-67 you highlighted that “Previously, we proposed that auxin and brassinolide may play important roles in giant pumpkin”, was the study on pumpkins, and if so, briefly mention the affected parameters?
Response 5: The sentence was rewritten:
Line 68-71: Previously, we found that the SNPs alterations in the IAA, BR and ER pathways may improve fruit enlargement, and proposed that auxin and brassinolide may play important roles in giant pumpkin fruit growth [18].
- Materials and methods
Plant materials and cultivation
Point 6: How did you arrive at the amount of fertilizer (organic and inorganic) to apply to the pumpkin plants, are they standard fertilizers? If too much was applied, how sure you are that the effects observed were a result of α-naphthaleneacetic acid (NAA) and 24-Epibrassinolide (EBR)?
Response 6: The amount of fertilizer applied to pumpkin plants in all treatments was the same, so as to avoid the impact of fertilizer on fruit growth.
Point 7: There is T10 in both tables 1 and 2, how is that possible?
Response 7: T10 was removed in table 2.
Point 8: It is hard to understand table 2, please make it easy to understand the information presented in it.
Response 8: The Table 2 and the note were revised, and the “4.2.2. NAA and EBR period and frequency of application” was rewritten:
Line 350-354: To further elucidate the beneficial effect of NAA and EBR on giant pumpkin fruit-weight, the combination treatment producing the largest single-fruit weight in Table 1 was selected, and then applied at different stages with different times during pumpkin growth. The specific design for NAA and EBR application periods and times are listed in Table 2.
Point 9: In line 387, fix the °C symbol
Response 9: Line 408, the °C symbol was added.
Point 10: In line 396, there must be a space between with and 50%
Response 10: Line 417, the space was added between with and 50%.
- Results
Point 11: Line 76- I don’t think it’s appropriate to start a new line with ‘except’ after a full stop
Response 11: Line 80, the ‘except’ was removed.
Point 12: Line 85- ‘’ fruit Fw’’ there is something wrong here, please fix, same in line 88…check throughout
Response 12: Line 89, the ‘’fruit Fw’’ was changed into ‘’Fw’’.
Point 13: Line 160- remove a full stop after %
Response 13: Line 167-, the sentence after “%” has been removed.
Point 14: As from section 2.2. why were the effects only referred to T18
Response 14: T18 was the most obvious combination to increase pumpkin fruit weight. We selected this combination to analyze the potential mechanism of NAA+EBR to increase pumpkin fruit weight.
Point 15: Overall, even though you presented enough data, it is however not easy for the reader to follow your results, try to be more precise and mention the most important results. In addition, avoid discussing your results in this section, only describe them.
Response 15: Following the reviewer’s advice, we checked and revised the manuscript to avoid the above problems.
- Discussion
Point 16: Line 273- …it also promoted….
Response 16: Line 286, the “also” was added.

Reviewer 4 Report
Research on the use of growth regulators to increase the size of crop fruits has been going on for a long time, but these studies remain relevant today. Moreover, the article presents the material of studies on the action of physiologically active substances at the genomic level, the metabolism of substances in the process of growth and ripening of fruits is studied. On the whole, this ensures both the relevance and scientific significance of the article, which, in terms of content, corresponds to the subject matter of the special issue of the journal and can be published in it.
In general, the article is clearly structured, the materials and methods are naturally connected with the results obtained and the discussion. However, some wishes can be expressed to improve the presented material.
There are misspellings: line 17 and 31 - you need α-naphthaleneacetic acid.
In conclusion, it would be desirable to conclude when, at what time, at what concentration it is necessary to use a substance or a combination of substances to obtain the maximum size of the fruit. From a practical point of view, it would be possible to recommend the technology of using physiologically active substances. There is no clear analysis of what is good or not high or low content of certain substances in fruits. It would be interesting to discuss in more detail the metabolism of substances in connection with the content of endogenous and exogenous physiologically active substances, the regulation of photosynthetic activity by the proposed scheme for the use of physiologically active substances.

Author Response
Response to Reviewer 4 Comments
Point 1: There are misspellings: line 17 and 31 - you need α-naphthaleneacetic acid.
Response 1: We added the α in line 17 and 31, and used the spell checker to examine the text.
Point 2: In conclusion, it would be desirable to conclude when, at what time, at what concentration it is necessary to use a substance or a combination of substances to obtain the maximum size of the fruit. From a practical point of view, it would be possible to recommend the technology of using physiologically active substances. There is no clear analysis of what is good or not high or low content of certain substances in fruits. It would be interesting to discuss in more detail the metabolism of substances in connection with the content of endogenous and exogenous physiologically active substances, the regulation of photosynthetic activity by the proposed scheme for the use of physiologically active substances.
Response 2: Following the reviewer’s advice, First, the conclusion section was rewritten:
Line 317-322: Taken together, exogenous NAA + EBR increased pumpkin fruit weight via an all-around coordinated action involving the source (leaf), transport (peduncle phloem sap), and the sink (fruit). Application of NAA + EBR can improve the photosynthetic capacity of leaves (sources), promote the synthesis of assimilates and water supply, enhance the transport efficiency of assimilates from the peduncle phloem sap to the growing fruit, and also improve the sink strength, which was conducive to the unloading of assimilates, thus providing carbon skeletons and energy for fruit growth, ultimately promoting the increase in pumpkin fruit weight. According to the results, the recommended combination of growth regulators for pumpkin production were 20 mg∙L-1 NAA + 1.0 mg∙L-1 EBR, and the optimal periods of application was spraying on the 20 days after planting, 0 DPA, 10 DPA and 20 DPA.
In addition, the following sentence was added at the “3. Discussion”:
Line 235-239: Effectively improve of leaf photosynthetic performance by exogenous application of NAA may be related to the increase of leaf area and chlorophyll content, the improvementof stomatal and non-stomatal restrictions, as well as the enhancement of photosynthetic system II [20,21].
Line 273-275: The effects of ETR treatment on photosynthetic capacity might be mediated by increased chlorophyll content and regulated of photosynthetic enzymatic activity [8,33-35].
Line 312-316: Although this study did not analyze the residues of NAA and EBR in pumpkins, a large number of previous studies showed that NAA dissipated rapidly in plants, and the residues of NAA were lower than the safety limit of pesticides in agricultural products and food (maximum residue limit, MRL), and EBR residues could not be detected in the plants which EBR was widely used [40-44].

Round 2
Reviewer 3 Report
Even though the authors did not highlight the changes they made in the manuscript, I believe that the manuscript significantly improved. The authors need to address a few issues before the manuscript can be accepted for publication. For a fast review process, can the authors highlight in bold the changes they made?
Line 352, "was selected" is repeated twice
In your results, either stick to fw or fruit weight, as long as you are consistent. ie: Line 96: "The single fruit Fw"_ this needs to be fixed, please check throughout the manuscript.
I am still failing to understand what T10 is doing in figure 2A, I advised the authors to remove it in Table 2 which has been revised but in figure 2A it is still there.
Line 146- "Treatment T18" Fix this
Line 237- "improvementof" there is no space???
Author Response
Point 1: Line 352, "was selected" is repeated twice
Response 1: Line 352, "was selected" was removed.
Point 2: In your results, either stick to fw or fruit weight, as long as you are consistent. ie: Line 96: "The single fruit Fw"_ this needs to be fixed, please check throughout the manuscript.
Response 2: The ‘’Fw’’ in the was changed into ‘’fruit weight’’.
Point 3: I am still failing to understand what T10 is doing in figure 2A, I advised the authors to remove it in Table 2 which has been revised but in figure 2A it is still there.
Response 3: In the experiment of the effect of different concentrations of NAA and EBR on giant pumpkin-fruit weight, T10 led to the highest fruit weight in all the combinations. Combination T10 represented that the giant pumpkin leaves and fruit were sprayed once with 20 mg∙L-1 NAA + 1.0 mg∙L-1 EBR at the rapid fruit expansion period (20 DPA). To further elucidate the beneficial effect of NAA and EBR on giant pumpkin fruit-weight, we carried out further experiments to study the effect of the optimal combination T10 on the weight of pumpkin fruit at different application periods and time, T16, T17, T18 and T19 represent the result of spraying twice, three times, four times and five times respectively of 20 mg∙L-1 NAA + 1.0 mg∙L-1 EBR. T10 was the result of spraying once of NAA + EBR, so it appears in Figure 2A in the revisions, and the revisions were named “ijms-1949009-Manuscript Revision (T10 in figure 2A) with track changes”and “ijms-1949009-Manuscript Revision (T10 in figure 2A)”.
At the same time, we also provide a version that removed the T10 in Figure 2 for final confirmation by the editor and reviewer in the revisions, and the revisions were named “ijms-1949009-Manuscript Revision (T10 remove in figure 2A) with track changes” and ijms-1949009-Manuscript Revision (remove T10 in figure 2A)”.
Point 4: Line 146- "Treatment T18" Fix this
Response 4: Line 146- ‘’Treatment T18’’ was changed into ‘’NAA+EBR combined treatment’’.
Point 5: Line 237- "improvementof" there is no space???
Response 5: Line 237- “the space” was added between “improvement” and “of”.
